# Cell-Type-Specific Effects of the Ovarian Cancer G-Protein Coupled Receptor (OGR1) on Inflammation and Fibrosis; Potential Implications for Idiopathic Pulmonary Fibrosis

**DOI:** 10.3390/cells11162540

**Published:** 2022-08-16

**Authors:** David J. Nagel, Ashley R. Rackow, Wei-Yao Ku, Tyler J. Bell, Patricia J. Sime, Robert Matthew Kottmann

**Affiliations:** 1Department of Medicine, Division of Pulmonary and Critical Care Medicine, University of Rochester Medical Center, Rochester, NY 14642, USA; 2Laboratory Medicine, Department of Pathology, Division of Clinical Chemistry, Johns Hopkins University, Baltimore, MD 21287, USA; 3BMW of North America, Woodcliff Lake, NJ 07675, USA; 4Department of Environmental Medicine, School of Medicine and Dentistry, University of Rochester, Rochester, NY 14642, USA; 5Department of Medicine, Virginia Commonwealth University Health System, Richmond, VA 23298, USA

**Keywords:** pulmonary fibrosis, OGR1 (GPR68), fibroblast, epithelial cell, GPCR, TGFβ signaling, epithelial to mesenchymal transition (EMT), myofibroblast differentiation

## Abstract

Idiopathic pulmonary fibrosis (IPF) is a disease characterized by irreversible lung scarring. The pathophysiology is not fully understood, but the working hypothesis postulates that a combination of epithelial injury and myofibroblast differentiation drives progressive pulmonary fibrosis. We previously demonstrated that a reduction in extracellular pH activates latent TGF-β1, and that TGF-β1 then drives its own activation, creating a feed-forward mechanism that propagates myofibroblast differentiation. Given the important roles of extracellular pH in the progression of pulmonary fibrosis, we sought to identify whether pH mediates other cellular phenotypes independent of TGF-β1. Proton-sensing G-protein coupled receptors are activated by acidic environments, but their role in fibrosis has not been studied. Here, we report that the Ovarian Cancer G-Protein Coupled Receptor1 (OGR1 or GPR68) has dual roles in both promoting and mitigating pulmonary fibrosis. We demonstrate that OGR1 protein expression is significantly reduced in lung tissue from patients with IPF and that TGF-β1 decreases OGR1 expression. In fibroblasts, OGR1 inhibits myofibroblast differentiation and does not contribute to inflammation. However, in epithelial cells, OGR1 promotes epithelial to mesenchymal transition (EMT) and inflammation. We then demonstrate that sub-cellular localization and alternative signaling pathways may be responsible for the differential effect of OGR1 in each cell type. Our results suggest that strategies to selectively target OGR1 expression may represent a novel therapeutic strategy for pulmonary fibrosis.

## 1. Introduction

Idiopathic pulmonary fibrosis (IPF) is an unrelenting fibrosing interstitial pneumonia with a median survival of approximately 3 years from the time of diagnosis [1] that affects 0.33–4.51 per 10,000 persons worldwide [2]. In 2014, two new medications were approved for the treatment of IPF (nintedanib and pirfenidone). These medications slow the rate of decline of some lung function parameters but do not affect patient-centered outcomes [3,4]. At present, the only curative treatment is lung transplantation. Unfortunately, not all patients are eligible for transplant and post-transplant complications pose a substantial burden. Therefore, IPF represents a large unmet medical need and novel therapies are needed.

The underlying mechanisms leading to the development of pulmonary fibrosis remain incompletely understood. One process involves injury to airway epithelial cells and subsequent transforming growth factor-beta1 (TGF-β1)-induced epithelial to mesenchymal transition (EMT). Epithelial cells that undergo EMT lose their polarity and demonstrate increased migration and invasion, resulting in an abnormal repair response [5]. This abnormal epithelial behavior signals adjacent fibroblasts to differentiate into myofibroblasts [6], and promote extracellular matrix remodeling [7,8,9,10]. TGF-β1 is the predominant pro-fibrotic cytokine responsible for the induction of myofibroblast differentiation [11,12,13]. TGF-β1 is produced as a latent protein that requires activation through dissociation from the latency associated peptide (LAP) [14]. Activation of TGF-β1 occurs through enzymatic degradation of LAP, mechanical strain, interactions with alpha integrins, and changes in pH and temperature [15,16,17,18,19]. We have demonstrated that TGF-β1 induces fibroblasts to generate excess lactic acid via lactate dehydrogenase A (LDHA), which reduces extracellular pH to an average of 6.7. This physiologic change in pH activates latent TGF-β1, which enhances additional myofibroblast differentiation [20].

We hypothesized that other pH-dependent processes may regulate the development of pulmonary fibrosis. One potentially important pathway involves a family of proton-sensing G-protein Coupled Receptors (GPCR). These GPCRs were initially identified in the early 2000s and are now recognized to contribute to the pathobiology of conditions ranging from malignancy [21,22] to asthma [23,24] and inflammatory bowel disease [25,26]. However, little is known about the role of proton-sensing GPCRs in fibrosis. There are four members within this family of receptors, the Ovarian Cancer G-Protein Coupled Receptor Protein1 (OGR1, GPR68), G-protein Coupled Receptor 4 (GPR4), G2 accumulation (G2A, GPR132), and T Cell Death-Associated Gene 8 (TDAG8, GPR65) [27,28,29,30]. These receptors sense extracellular acidification via protonation of histidine residues located on the extracellular domain of the receptor [28,31]. OGR1 was originally cloned from an ovarian cancer cell line [32] but has subsequently been found to be expressed in the spleen, testis, small intestine, kidney, brain, heart, and lung [27]. Although sphingosylphosphorylcholine, galactosylsphingosine, and lysophosphatidylcholine have been proposed to be endogenous ligands, the validity of these findings remains in question and the role of OGR1-ligand interactions remains largely unknown [33,34].

OGR1 is inactive at a pH of 7.8 but fully active at a pH of 6.8 [28], where it exerts ligand-independent constitutive activity [35]. In vitro expression of this receptor has been shown to suppress metastasis in prostate cancer [35], mitigate cell migration in breast cancer [36], and regulate acid-induced apoptosis in endplate chondrocytes [37]. However, in other studies, the absence or reduction of OGR1 expression inhibited melanoma tumorigenesis [38], was protective against inflammation in an IBD mouse model [39], and protected against the development of murine IBD-associated fibrosis [40]. These disparate functions may be explained by the observation that OGR1 may signal through Gαs and Gαq signaling pathways [24] and/or that OGR1 displays biased agonism within the same medication class [41]. Thus, the role of OGR1 in human disease is complex and appears to be context- and tissue-dependent.

We sought to determine if OGR1 expression was dysregulated in pulmonary fibrosis and if alterations in OGR1 expression modulate EMT and myofibroblast differentiation. Our data demonstrate that OGR1 protein expression is downregulated in pulmonary fibrosis and that OGR1 negatively regulates pro-fibrotic signaling in lung fibroblasts. However, in epithelial cells, OGR1 is pro-inflammatory and promotes EMT. We also demonstrate that OGR1 appears to use different signaling pathways and reside in different subcellular compartments in the epithelium versus fibroblasts. These findings warrant further investigation into the feasibility of using precision targeting of OGR1 as an anti-fibrotic therapy.

## 2. Materials and Methods

### 2.1. Human Lung Biopsy Samples

Lung tissue samples were obtained from patients using existing URMC RSRB approved human subject protocols and the Lung Tissue Research Consortium. All tissue was de-identified and collected in a standard manner according to approved institutional review board protocols.

### 2.2. Primary Fibroblast Culture

Human lung fibroblasts were obtained from explanted tissues from healthy controls (people undergoing lung biopsy for non-ILD related conditions) and those getting biopsy to confirm the diagnosis of IPF as previously described [42]. All donors gave written consent, patient information was de-identified, and all protocols followed URMC Institutional Review Board guidelines. Cells were cultured in Modified Eagle Medium (Gibco, Waltham, MA, USA) supplemented with 10% fetal bovine serum (Sigma-Aldrich, St. Louis, MO, USA), 1% L-glutamine, and 1% antibiotic–antimycotic reagents (Gibco). Passages from 3 to 9 were used for all experiments. For epithelial cell experiments, 16-HBE cells were utilized under identical conditions.

OGR1 over-expression was accomplished using the X-treme Gene HP protocol (Roche, Basel, Switzerland) and an OGR1 plasmid previously described [43]. Fibroblasts were plated at a density of 7 × 10^4^ cells/well in MEM with serum, and plasmid DNA was delivered in serum-free OptiMEM (Gibco, Fisher Scientific, Waltham, MA, USA). After overnight incubation, cells were administered TGF-β1 (1 ng/mL, R&D Systems, Minneapolis, MN, USA), CTGF (0.5 µL/mL, R&D Systems), LPS (0.02 µL/mL, Fisher Scientific), or DMSO (in equal volume, Sigma-Aldrich) in fresh MEM for an additional 48 h.

OGR1 expression was knocked down in human lung fibroblasts with SMART pool ON-TARGET plus GPR68 siRNA and control conditions utilizing non-targeting SiRNA (Dharmacon, Lafayette, CO, USA) with the X-treme Gene SiRNA reagent per manufacturer instructions (Roche, Basel, Switzerland). The following morning, cells were treated with TGF-β1 or DMSO in fresh serum-containing MEM media for 48 h.

### 2.3. Quantitative Real-Time Polymerase Chain Reaction

Total RNA was isolated using Trizol (Invitrogen, Carlsbad, CA, USA). Reverse transcription utilized the iScript cDNA synthesis kit (Bio-Rad, Hercules, CA, USA). Real-time PCR reactions were performed with SYBR Green (Bio-Rad) and were analyzed with a T100 Thermal Cycler (Bio-Rad). The following sequences were used for primers: housekeeping gene (**18S**) forward: 5′GCTTGCTCGCGCTTCCTTACCT, reverse: 5′TCACTGTACCGGCCGTGCGTA; **OGR1** forward: 5′TGTACCATCGACCATACCATCC, reverse: 5′GGTAGCCGAAGTAGAGGGACA.

**Collagen 1a1** forward: 5′CTGCTGGCAAAGATGGAG, reverse: 5′ACCAGGAAGACCCTGGAATC; **Collagen 3a1** forward: 5′AAATGGCATCCCAGGAG, reverse: 5′ATCTCGGCCAGGTTCTC; **α-Smooth muscle actin** forward: 5′GTGTTGCCCCTGAAGAGCAT3′, reverse: 5′GCTGGGACATTGAAAGTCTCA3′; **Fibronectin** forward: 5′TTGAAGGAGGATGTTCCCATCT3′, reverse: ACAGACACATATTTGGCATGGTT3′.

### 2.4. Western Blotting

Cellular or whole lung lysates were run on a 10% SDS-PAGE gel and transferred using the Trans-Blot Turbo Transfer System (Bio-Rad). Low fluorescence polyvinylidene difluoride (PVDF) membranes (EMD Millipore, Billerica, MA, USA) were activated in methanol and blocked with EveryBlot Blocking Buffer (Bio-Rad) followed by incubation with the following primary antibodies: OGR1 (“GPR68” Invitrogen, Carlsbad, CA, USA), beta-tubulin (Abcam, Cambridge, MA, USA), α-SMA (Sigma-Aldrich, St. Louis, MO, USA), phospho-Smad2/total Smad2 (Abcam), and Col1A1 (Aviva Systems Biology, San Diego, CA, USA). Immunofluorescent secondary antibodies (StarBright520 for mouse and StarBright700 for rabbit, Bio-Rad) were diluted at 1:20k in 5% milk and TBST and membranes were incubated for one hour. Membranes were imaged with a Chemidoc MP fluorescent imager (Bio-Rad) and quantified with Image Lab (version 6.1, Bio-Rad).

### 2.5. Subcellular Fractionation

This protocol was adapted from the Thermo Fisher Protocol for “Subcellular Protein Fractionation of Cultured Cells”. Briefly, 1 mL of trypsin was added to the culture plate, followed by 1 mL of MEM. Cell suspensions were retrieved and placed in microcentrifuge tubes and spun at 500× *g* at 4 degrees Celsius for 5 min. The supernatant was removed, samples were rinsed with 1 mL cold PBS, and re-centrifuged at 500× *g* for 8 min. Supernatant PBS was subsequently removed and 200 µL of “CEB” buffer, with protease inhibitor, was added, and samples were gently mixed on a rocker for 10 min at 4 °C. Samples were centrifuged at 500× *g* for 5 min, and the supernatant was transferred to a sample tube (cytoplasmic fraction). Then, “MEB” buffer with protease inhibitor was added to the remnant samples. Incubation for 10 min at 4 °C occurred again, followed by centrifugation at 3000× *g* for 5 min. The supernatant was transferred to a clean tube (membrane fraction) and “NEB” (with protease inhibitor) buffer was added and incubated for 10 min at room temperature. Samples were then vortexed and placed on the rocker for 30 min at 4 °C. Samples were then centrifuged at 5000× *g* for 5 min and the supernatant was transferred to a clean microcentrifuge tube (soluble nuclear fraction). Finally, CaCl_2_ and MN were added to the NEB buffer; remnant samples were incubated for 5 min at 37 °C in a water bath. Samples were then centrifuged at 16,300× *g* for 5 min at 4 °C. The supernatant was removed and transferred (chromatin-bound nuclear extract). Samples were then run according to the western blot protocol detailed above. Subcellular fractions were verified via antibodies to HSP90 (cytoplasmic fraction, Cell Signaling Technology, Danvers, MA, USA), Gβ (membrane fraction, Santa Cruz), AIF (mitochondrial membrane and nucleus, Cell Signaling Technology), lamin A/C (nuclear membrane, Cell Signaling Technology), and histone H3 (chromatin-bound, Cell Signaling Technology).

### 2.6. IL-6 ELISA

The Human IL-6 DuoSet ELISA (R&D systems) analysis was performed according to manufacturer’s instructions. Briefly, diluted Capture Antibody was applied to a 96 well-plate and incubated overnight. The following morning, each well was aspirated and washed with wash buffer for a total of three times. Wells were then blocked with Reagent Diluent for 1 h at room temperature. Then, each well was aspirated and washed three times with wash buffer. Next, 100 µL of sample or standard was added to each well and incubated for 2 h at room temperature. Wells were then aspirated and washed three times. Then, 100 µL of Detection Antibody was added to each well and incubated for two hours at room temperature. Wells were then aspirated and washed as above. Then, 100 µL of Streptavidin-HRP (working dilution) was added to each well and incubated for 20 min at room temperature. Wells were then aspirated and washed again, and 100 µL of Substrate Solution was added to each well for 20 min at room temperature. Stop Solution was then added to each well and the plate was gently tapped to ensure mixing. An iMark microplate reader (Bio-Rad) was then used to measure absorbance at 540 nM.

### 2.7. Data Analysis

Data is presented as mean values +/− standard error of the mean. All experiments were performed in triplicate at a minimum. Statistical analysis using one-way analysis of variance (ANOVA) with Tukey multiple comparison method or unpaired *t*-tests were performed with Graph Pad Prism version 9 (San Diego, CA, USA). A *p* value of <0.05 was considered statistically significant.

## 3. Results

### 3.1. OGR1 Expression Is Down-Regulated in Lung Tissue from Patients with Idiopathic Pulmonary Fibrosis

OGR1 is highly expressed in whole lung lysates from healthy patients as detected by western blot analysis (Figure 1A). However, in patients with a diagnosis of IPF, OGR1 protein expression is heterogeneously reduced (Figure 1A). OGR1 protein expression is also significantly reduced in fibroblasts isolated from patients with IPF compared with healthy controls (Figure 1B). Interestingly, when we analyzed the expression of OGR1 mRNA among healthy and IPF human fibroblasts, we found that mRNA levels did not differ at baseline (Figure 1C). This finding was consistent with OGR1 expression profiles in the IPF cell atlas database (http://www.ipfcellatlas.com/) [44]. However, when fibroblasts were treated with TGF-β1, IPF fibroblasts were more sensitive to OGR1 downregulation (Figure 1C). These results suggest that OGR1 protein expression is down-regulated post-translationally in pulmonary fibrosis, and that this is regulated in a TGF-β1-dependent manner.

### 3.2. OGR1 Downregulation Induces Myofibroblast Differentiation in Human Lung Fibroblasts

As OGR1 protein is downregulated in IPF lung tissue and fibroblasts, we hypothesized that a decrease in OGR1 expression may induce a fibrotic phenotype even in the absence of TGF-β1 stimulation. To assess this hypothesis, OGR1 expression was knocked down via siRNA in healthy human fibroblasts. We observed an approximate 80% reduction in OGR1 mRNA expression after siRNA administration (Figure 2A). Decreasing OGR1 expression induced significant increases in expression of markers of myofibroblast differentiation including collagen 1A1 (Figure 2B), collagen 3A1 (Figure 2C), alpha smooth muscle actin (Figure 2D), and fibronectin (Figure 2E). These data suggest that decreased OGR1 expression is sufficient to induce myofibroblast differentiation and that OGR1 expression alone is an important negative regulator of myofibroblast differentiation.

### 3.3. Fibroblast OGR1 Expression Is Down-Regulated by TGF-β1 but Not LPS or CTGF

To further examine the specificity of TGF-β1 mediated downregulation of OGR1, we examined if another cytokine, connective tissue growth factor (CTGF), which promotes fibrosis via enhanced mesenchymal chemotaxis, proliferation, and collagen synthesis [45], also downregulated OGR1 protein expression. We also examined the effects of a potent pro-inflammatory/fibrotic stimulant, lipopolysaccharide (LPS), on OGR1 protein expression. As expected, in the presence of TGF-β1, OGR1 protein expression is significantly decreased (Figure 3A). However, neither CTGF (Figure 3B) nor LPS (Figure 3C) significantly altered fibroblast OGR1 protein expression. These results demonstrate that fibroblast-specific downregulation of OGR1 is primarily mediated by TGF-β1 and therefore may offer additional insight into the regulation of OGR1 expression.

### 3.4. OGR1 Inhibits TGF-β1 Induced Myofibroblast Differentiation

We next hypothesized that OGR1 serves as a negative regulator of TGF-β1mediated signaling. To test this hypothesis, we over-expressed OGR1 in both healthy and IPF-derived fibroblasts. A green fluorescence protein (GFP) plasmid was used as a control. (Figure 4A,B,E,F). We subsequently assessed for changes in markers of myofibroblast differentiation. Under basal conditions, OGR1 overexpression had no effect on α-SMA (Figure 4A,C) or collagen expression (Figure 4D). As expected, treatment with TGF-β1 significantly increased α-SMA and collagen protein expression (Figure 4A,C,D). In IPF-derived fibroblasts, similar changes were observed (Figure 4E,G,H). However, OGR1 overexpression also reduced basal expression of collagen in IPF fibroblasts (Figure 4E,H).

We next examined the effects of knocking down OGR1 protein. Reductions of OGR1 were significantly reduced compared to a non-targeting siRNA (Figure 5A,B,E,F). In contrast to what was observed in healthy fibroblasts, reducing OGR1 expression did cause myofibroblast differentiation under basal conditions, though it did so in a different manner in healthy versus IPF fibroblasts (Figure 5C). In healthy fibroblasts, OGR1 siRNA increased α-SMA expression (Figure 5C) but did not have a significant effect on collagen expression. However, similar changes were not observed with IPF-derived fibroblasts: knock-down of basal OGR1 expression caused a significant increase in collagen expression (Figure 5H) but not α-SMA expression (Figure 5G). In both healthy and IPF fibroblasts, TGF-β1 induced myofibroblast differentiation, but reducing OGR1 expression had no significant additive effect (Figure 5A,C,E,G). These results demonstrate that OGR1 negatively regulates TGF-β1-induced myofibroblast differentiation and suggests that maintenance of fibroblast-specific OGR1 expression and/or activity may be a novel anti-fibrotic strategy.

### 3.5. Epithelial OGR1 Promotes EMT

Due to the significant contribution of epithelial dysfunction to pulmonary fibrosis and the existing literature, we next examined the effects of manipulating OGR1 protein expression in epithelial cells. Based on the literature, we hypothesized that OGR1 would be pro-inflammatory in epithelial cells, though there was little data to suggest if OGR1 would affect fibrosis-related outcomes in the epithelium. To assess both fibrotic and inflammatory changes in the epithelium, we overexpressed OGR1 in 16-HBE (Figure 6A) and assessed changes in protein expression that are hallmarks of EMT. Increased expression of OGR1 caused increased levels of collagen (Figure 6B) and fibroblast-specific protein 1 (FSP1) (Figure 6C) but decreased expression of E-cadherin (Figure 6D). These changes are all typical of EMT and support previous data suggesting that OGR1 may be both pro-fibrotic and pro-inflammatory in epithelial cells.

### 3.6. OGR1 Promotes Epithelial Inflammation but Does Not Induce Fibroblast Inflammation

We then examined a simple marker of cellular inflammation by measuring in vitro concentrations of secreted IL-6 in the presence of overexpressed OGR1 and/or known pro-inflammatory cytokines. In epithelial cells, OGR1 overexpression alone was sufficient to induce a significant amount of IL-6 secretion and was just as potent as TGF-β1, LPS, and CTGF (Figure 7A). Interestingly, increased OGR1 expression did not have any additive or synergistic effect on these cytokines. This either suggests that OGR1 utilizes the same mechanism of action as the growth factors, or that there is a ceiling effect with relation to IL-6 production. In contrast, OGR1 overexpression in fibroblasts did not induce IL-6 secretion (Figure 7B). There was also no effect of OGR1 overexpression on the ability of TGF-β, LPS, or CTGF to induce inflammation. These data suggest that OGR1 is sufficient to induce inflammation in epithelial cells, but in fibroblasts, it neither promotes nor mitigates inflammation.

### 3.7. OGR1 Inhibits TGF-β1 Induced Smad2 Phosphorylation, but Only in Fibroblasts

Given the dissimilarity in the downstream effects of OGR1 in epithelial versus fibroblasts, we next turned our attention to cell-type-specific differences that might explain the disparate actions of OGR1. To this end, we examined the effect of OGR1 overexpression on Smad2 signaling, a canonical effector of TGF-β1. Activation of the TGF-β1 receptor induces phosphorylation of Smad2 and Smad3. Activated Smads then form a trimer with Smad4, and this complex translocates to the nucleus to either stimulate or repress various gene transcription targets, ultimately inducing EMT [46]. As expected, TGF-β1 induced significant Smad2 phosphorylation in epithelial cells (Figure 8A). There was no effect, under basal conditions, on Smad phosphorylation when OGR1 was overexpressed. In the presence of TGF-β1, OGR1 overexpression did not inhibit Smad2 phosphorylation (Figure 8A). This suggests that epithelial canonical TGF-β1 signaling is independent of OGR1.

We then examined the effects of OGR1 overexpression in fibroblasts. Similar to epithelial cells, TGF-β1 induced significant Smad2 phosphorylation (Figure 8B). Again, OGR1 overexpression alone had no effect on basal Smad2 phosphorylation. However, in the presence of TGF-β1, OGR1 overexpression significantly reduced Smad2 phosphorylation back to baseline levels (Figure 8B). These data demonstrate that OGR1 has differential effects on canonical TGF-β1 signaling that is dependent on the cell type. Future investigation should examine the mechanism(s) by which OGR1 inhibits TGF-β1 signaling, how OGR1 promotes epithelial inflammation, and if OGR1 negatively regulates non-canonical TGF-β1 signaling pathways.

### 3.8. OGR1 Displays Cell-Type Specific Subcellular Location

To further characterize potential cell specific OGR1 signaling, we fractioned cells into standard subcellular compartments (cytoplasm, membrane, nuclear, and chromatin-bound) in both epithelial cells and fibroblasts. The various compartments were verified using markers that are known to be associated with a given fraction (Figure 9B). In epithelial cells, OGR1 was highly expressed in the cytoplasm, nuclear, and chromatin-bound fractions (Figure 9A, lower panel). This suggests OGR1 may regulate transcription as well as other behaviors in epithelial cells on the basis of its subcellular location. In contrast, OGR1 was only detected in the nuclear fraction of fibroblasts (Figure 9A, upper panel). The discovery of OGR1 in the nuclear fraction was surprising, and whether the entire receptor is capable of translocation, or whether it is a cleavage product of OGR1 that translocates, is not yet clear. The cell-type-specific localizations may offer an additional explanation as to why the same receptor may have distinct functions across cell types. Although there is increasing recognition of the existence of nuclear-bound GPCRs [47], classic teachings would have hypothesized that OGR1 would be in the membrane fraction. This unique property may confer an advantage to exploit in terms of developing cell- and subcellular fraction-specific pharmacological therapies.

## 4. Discussion

Chronic fibrotic pulmonary diseases convey a high morbidity and mortality [1,48], and an individual’s response to therapy varies considerably [49]. Therefore, a more in-depth understanding of the pathogenesis and efficacious anti-fibrotic therapies remain a critical unmet medical need. The pathologic hallmark of IPF is the fibroblast foci, areas that are enriched with activated myofibroblasts that deposit excessive extracellular matrix and disrupt normal lung architecture [9]. Myofibroblasts have a hybrid phenotype between smooth muscle cells and fibroblasts, demonstrated by their ability to express contractile proteins like α-smooth muscle actin [50]. This characteristic allows myofibroblasts to repair wounds, but for reasons that are not entirely understood, pulmonary fibrosis develops from an aberrant repair process. Epithelial cell injury and dysfunctional repair mechanisms also contribute to fibrosis [5]. Transforming growth factor-β1 promotes pathologic fibrosis in many diseases [51] via induction of EMT and myofibroblast differentiation [13]. We have previously demonstrated that TGF-β1 leads to acidification of the extracellular space [20,52]. This leads to increased LDHA expression and activity, which promotes activation of latent TGF-β1 and causes downstream fibrosis. In this manuscript, we sought to determine additional mechanisms by which changes in the extracellular pH are conveyed to the cell and how these signals may attenuate the pro-fibrotic effects of TGF-β1.

Here we expand the knowledge of how proton-sensing GPCRs, in particular the Ovarian Cancer G-Protein Coupled Receptor 1, negatively regulate pathologic fibrotic signaling. We provide several new insights for OGR1, including cell-type-specific signaling, the contradictory findings suggesting that OGR1 is both pro- and anti-inflammatory, and that OGR1 is unique in its subcellular location. There are several reports that demonstrate the spatial and temporal variability of OGR1 signaling and its ramifications [24,35,36,37,38,39,53]. For example, Matsuzaki et al. found that OGR1 increased connective tissue growth factor (CTGF) in airway smooth muscle cells in response to extracellular acidification. However, in a human fibroblast cell line (HFL-1) and a bronchial epithelial cell line (BEAS-2B), which both express OGR1 at higher levels than other members of this GPCR family, CTGF expression was unchanged in response to extracellular acidification [53]. There are also conflicting reports as to whether OGR1 promotes healthy responses [35,36,37] or pathologic ones [38,39,40]. OGR1 has also been shown to signal through both Gαq [28] and Gαs [54,55] G-proteins. However, it is not understood how, or in which contexts, OGR1 associates with different G-proteins. Finally, different agonists of OGR1 from the same medication class can elicit variable down-stream signaling [41]. These properties highlight the complexity of OGR1 signaling and reflect differences in cell types (i.e., epithelial versus fibroblasts), the adjacent cellular context (normal versus acidic environment), and the agonist being utilized (lorazepam versus sulazepam). We add to that complexity with this manuscript by demonstrating that OGR1 appears to have opposite functions in that it promotes inflammation in epithelial cells (and is pro-fibrotic), while in the adjacent fibroblast, it is anti-fibrotic and appears to be independent of inflammation. Here, we report that OGR1 protein expression is significantly decreased in whole-lung lysates and fibroblasts isolated from people with IPF (Figure 1A,B). While there is certainly a degree of heterogeneity of OGR1 expression in people with IPF, we found that explanted samples had lower levels of OGR1 relative to lung biopsy samples (data not shown). This raises the possibility that as lung fibrosis progresses over time, OGR1 expression may be increasingly downregulated. The IPF Cell Atlas has been a tremendous resource for researchers, and their data demonstrate that OGR1 mRNA expression was unchanged between healthy and fibrotic fibroblasts. Indeed, our data further extend these findings and suggest that a post-translational modification, receptor desensitization, or receptor downregulation may occur after mRNA transcription, leading to decreased protein expression. Additional investigation is required to further define the mechanism(s) of OGR1 protein downregulation.

We also demonstrate that in vitro OGR1 expression (both mRNA and protein) is negatively regulated by TGF-β1 (Figure 1 and Figure 3). In fibroblasts, OGR1 appears to be protective against myofibroblast differentiation. Specifically, knocking down OGR1 promotes myofibroblast/pro-fibrotic gene expression (Figure 2). In addition, overexpression of OGR1 attenuates TGF-β1 induced myofibroblast differentiation in both healthy and IPF-derived fibroblasts (Figure 4 and Figure 5). We propose that the ability of TGF-β1 to downregulate a counter-regulatory protein like OGR1 further enhances a pro-fibrotic feed forward loop. However, this is in direct contrast to Hutter et al. [40], who demonstrated that OGR1 promotes intestinal fibrosis. It is possible that OGR1 utilizes a different G-protein in the intestine compared to the lung. It is also possible that an individual stimulus confers bias to the interaction between the receptor and the G-protein (biased agonism). In addition, this manuscript only examines in vitro signaling, and it is likely that critical elements found in humans contribute to OGR1 regulation. These discrepant findings highlight the need for a better understanding of the complex proton sensing GPCR signaling pathways.

The role of inflammation in the development of pulmonary fibrosis remains controversial because pre-clinical models do not necessarily correlate with clinical observations. For example, transient expression of IL-β in rodents led to alveolar inflammation and interstitial fibrosis through increased concentrations of IL-6, tumor necrosis factor-α (TNF-β), and TGF-β1 and platelet derived growth factor [56]. In acute exacerbations of IPF, increased levels of IL-1β and TNF-α expression were seen in lung biopsy samples [57], but IPF is generally considered a non-inflammatory disease [58]. There are several lines of evidence demonstrating the pro-fibrotic effects of OGR1 in the epithelium [23,24,25,26,39]. Here, we offer data that further supports the pro-inflammatory nature of OGR1 in epithelium (Figure 7). Interestingly, we did not see the same pro-inflammatory property in fibroblasts. We also demonstrate that OGR1 has an additional way to promote fibrosis; by inducing EMT, a crucial profibrotic signal (Figure 6).

To better address this apparent contradiction, we go on to show that OGR1 does not signal through the canonical Smad-mediated signaling pathway in the epithelium (Figure 8). However, OGR1 negatively regulates Smad2 signaling in fibroblasts. Again, this suggests a cell-type-specific nature to downstream OGR1 signaling. We also demonstrate different subcellular localization of OGR1 in epithelial cells and fibroblasts (Figure 9). In 16-HBE and intestinal epithelial cells, OGR1 signals through Gq_11_/PLC/Ca^2+^ [26,59], and we recently demonstrated that a small-molecule positive allosteric modulator of OGR1, Ogerin, stimulates the Gα_s_/PKA pathway in dermal, orbital, and pulmonary fibroblasts (PONE-D-22-03653R1). PKA activation was further enhanced in fibroblasts by decreasing the pH from 7.4 to 6.8. This body of evidence supports the ability of OGR1 to differentially regulate secondary messengers in a cell-type-, pH-, and G-protein subunit-specific manner. Our data offer additional, though incomplete, insight into this intriguing family of orphan GPCRs, and we hope that inherent cell specific differences can be exploited in developing novel therapeutic targets.

There is a growing recognition that genetic variations play a role in the development and clinical course in idiopathic pulmonary fibrosis. However, only about 1/3 of IPF diagnoses are attributed to common genetic variants [60]. Therefore, at our present level of understanding, it does not appear that the majority of IPF cases can be explained by genetic alterations alone. It is interesting to note that the OGR1-deficient mouse does not develop an overt phenotype. This may be true because this mouse is only deficient in a portion of the receptor [38] rather than a complete receptor knock out. It is also possible that in vivo elimination is not sufficient to cause fibrosis without a traditional pro-fibrotic challenge, or that there is compensatory action by another receptor.

Our future work will further characterize the response of OGR1 knock-out mice to bleomycin challenge. We anticipate that they will be more susceptible to bleomycin injury, and we plan to further characterize the cell-specific responses of the OGR1-deficient mouse to bleomycin challenge. Due to the complexity of the disease, the idea of combinatorial therapy for IPF is on the rise. Single-therapy medications aimed at one target may not be enough to demonstrate a significant clinically meaningful outcome due to signal redundancy. Indeed, the current FDA-approved anti-fibrotic medications slow the progression of disease, but patient-reported symptoms remain unchanged. Pooled data from the INPULSIS and TOMORROW trials involving nintedanib demonstrate a reduction in acute exacerbations [61], and secondary endpoint analysis with pirfenidone and nintedanib suggests an improvement in all-cause and IPF-related mortality [3,62]. These additional benefits need to be confirmed with larger randomized clinical trials designed to assess this question. Given that OGR1 deficiency does not confer an overt pulmonary fibrotic phenotype by itself, we recognize that solely targeting OGR1 as a new anti-fibrotic strategy is misguided for several reasons. Rather, we suggest using a multimodal approach to IPF to combat the redundant nature of pro-fibrotic signaling pathways. Given that GPCRs are the most common entity targeted by FDA-approved drugs [63] and that extracellular pH is decreased in pulmonary fibrosis [20,52,64], OGR1 represents an exciting, though challenging, potential therapeutic target for IPF.

## Figures and Tables

**Figure 1 cells-11-02540-f001:**
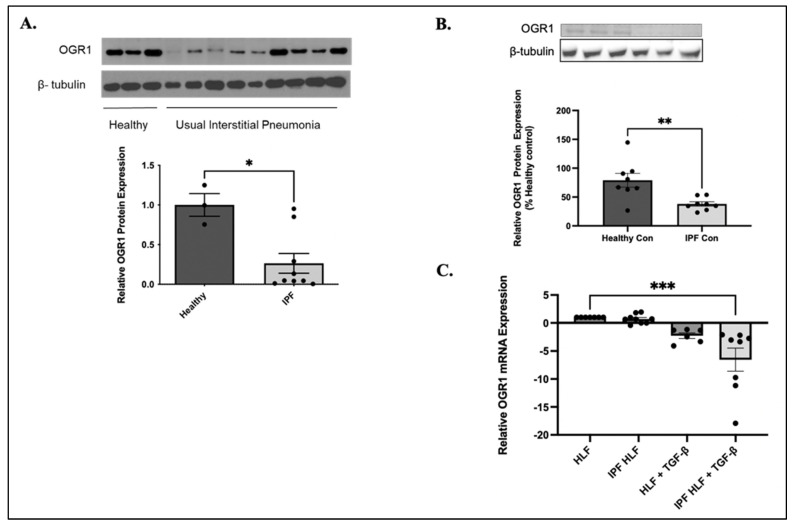
OGR1 protein expression, but not mRNA, are reduced in IPF. Lung tissues from healthy patients and those with IPF were analyzed for OGR1 protein expression by western blot. Densitometric analysis was performed and data represent mean OGR1 expression relative to β-tubulin ± SEM, * *p* = 0.0105 (**A**). We next compared OGR1 protein expression between healthy human lung fibroblasts and fibroblasts obtained from patients with IPF. Under basal conditions, IPF-derived fibroblasts demonstrated significantly decreased OGR1 protein expression, ** *p* = 0.0056 (**B**). We then examined expression of mRNA from healthy and IPF human fibroblasts via qRT-PCR; there was no difference in baseline OGR1 mRNA levels. However, IPF-derived fibroblasts were more sensitive to down-regulation by TGF-β, *** *p* = 0.0001 (**C**). Data represent mean expression ± SEM.

**Figure 2 cells-11-02540-f002:**
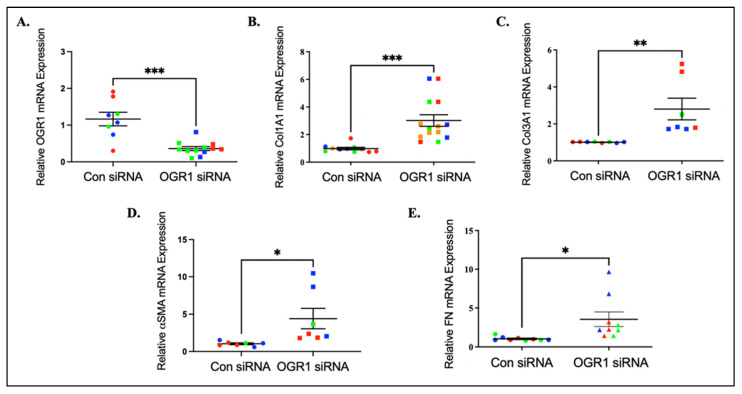
Knocking down OGR1 expression, through siRNA, causes pro-fibrotic gene expression. Healthy human lung fibroblasts were treated with non-targeting or OGR1 siRNA. The expression of OGR1 (**A**), collagen 1A1 (**B**), collagen 3A1 (**C**), α-SMA (**D**), and fibronectin (**E**) were subsequently assessed by qRT-PCR. Results are displayed as candidate mRNA expression relative to 18 S, each set of colors represent a unique cell line isolated from different donors. Data represent mean expression ± SEM (n = 3/treatment group, repeated in triplicate in three different cell lines). *** *p* = 0.0001 (**A**), *** *p* = 0.0003 (**B**), ** *p* = 0.0058 (**C**), * *p* = 0.0303 (**D**), * *p* = 0.0165 (**E**).

**Figure 3 cells-11-02540-f003:**
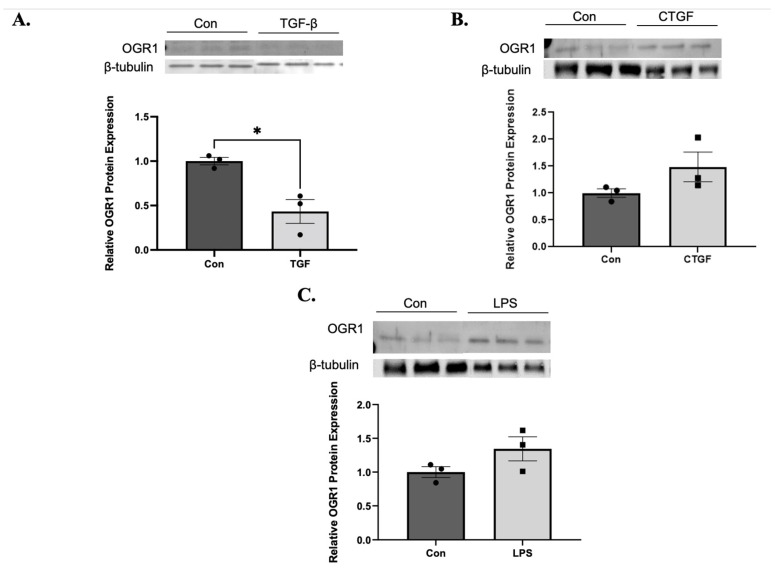
TGF-β, but not CTGF or LPS, reduce OGR1 protein expression. Fibroblasts derived from healthy subjects treated with TGF-β1 (1 ng/mL), CTGF (0.5 µL/mL), or LPS (0.02 µL/mL) for 48 h, and protein was then harvested. Western blotting was performed to assess for changes in OGR1 expression. As previously demonstrated, TGF-β1 caused a significant reduction in OGR1 expression (**A**). However, there were no significant differences when fibroblasts were treated with CTGF (**B**) or LPS (**C**). Densitometry analysis is included below each representative western blot and data represent mean expression ± SEM, * *p* = 0.0155.

**Figure 4 cells-11-02540-f004:**
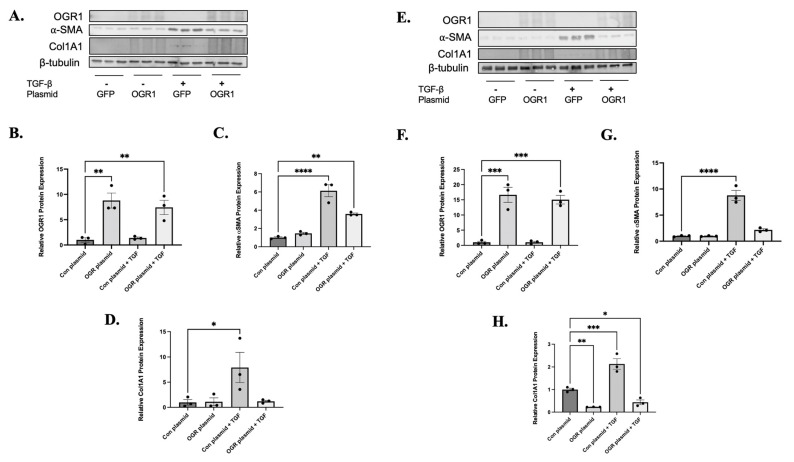
OGR1 overexpression reduces the ability of TGF-β to induce myofibroblast differentiation. Human fibroblasts (healthy: **A**–**D**, IPF: **E**–**H**) were transfected with a plasmid expression GFP (con) or OGR1 and were subsequently treated with either TGF-β1 or DMSO as a control. Protein lysates were then harvested and western blotting was performed with antibodies against OGR1, α-SMA, collagen 1A1, and beta-tubulin (as a loading control). Representative western blots are presented in (**A** and **E**, respectively), and densitometry analysis is graphed below. OGR1 plasmids were successfully expressed in both healthy (**B**, ** *p* = 0.0020 and 0.0065) and IPF-derived fibroblasts (**F**, *** *p* = 0.0002 and 0.0004). As expected, TGF-β1 significantly increased expression of α-SMA in both healthy (**C**, **** *p* < 0.0001) and IPF-derived fibroblasts (**G**, **** *p* < 0.0001). However, overexpression of OGR1 attenuated TGF-β1-induced α-SMA expression in healthy fibroblasts (**C**, ** *p* = 0.0017) and IPF fibroblasts (**G**, difference between con plasmid + TGF-β1 and OGR plasmid *p* = 0.0025 by Student’s *t*-test). Again, as expected, treatment with TGF-β1 caused increased collagen1A1 expression in healthy (**D**, * *p* = 0.0364) and IPF fibroblasts (**H**, *** *p* = 0.0008). Again, OGR1 overexpression led to significant reductions in TGF-β induced collagen in healthy (**D**, *p* = 0.009 by Student’s *t*-test) and IPF fibroblasts (**H**, * *p* = 0.0027 by student *t*-test). Interestingly, in IPF fibroblasts, OGR1 overexpression decreased collagen expression in an unstimulated state (**H**, ** *p* = 0.0084). Data represent mean expression ± SEM.

**Figure 5 cells-11-02540-f005:**
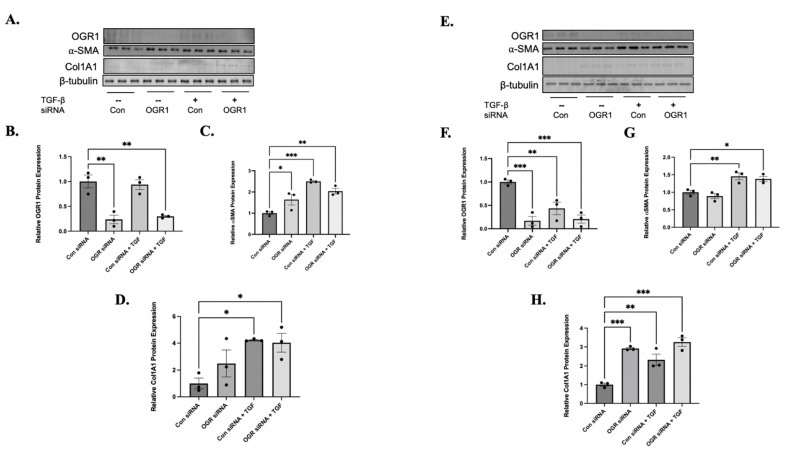
Knock-down of OGR1 expression does not have additive or synergistic effects on TGF-β induced myofibroblast differentiation. Human fibroblasts (healthy: **A**–**D**, IPF: **E**–**H**) were transfected with either non-targeting (con) or OGR1 siRNA and were subsequently treated with either TGF-β1 or DMSO as a control. Protein lysates were then harvested and western blotting was performed with antibodies against OGR1, α-SMA, collagen 1A1, and beta-tubulin (as a loading control). Representative western blots are presented (**A** and **E**, respectively), and densitometry analysis is graphed below. OGR1 was successfully knocked down in both healthy (**B**, ** *p* = 0.0010 and 0.0017) and IPF-derived fibroblasts (**F**, *** *p* = 0.0007 and 0.0009). As previously demonstrated, treatment with TGF-β1 decreased basal OGR1 expression in IPF-derived fibroblasts (** *p* = 0.0071). As expected, TGF-β1 significantly increased expression of α-SMA in both healthy (**C**, *** *p* = 0.0003 and ** *p* = 0.0028) and IPF-derived fibroblasts (**G**, ** *p* = 0.0076 and * *p* = 0.02). Interestingly, in healthy fibroblasts, decreasing basal expression of OGR1 caused a significant increase in α-SMA expression (**C**, * *p* = 0.0373). Again, as expected, treatment with TGF-β1 caused increased collagen1A1 expression in healthy (**D**, * *p* = 0.0364) and IPF fibroblasts (**H**, *** *p* = 0.0037 and ** *p* = 0.0001, respectively). In IPF fibroblasts, decreasing basal expression of OGR1 caused increased collagen1A1 expression (**H**, *** *p* = 0.0003). Suppressing OGR1 expression did not add to the ability of TGF-β to induced myofibroblast differentiation. Data represent mean expression ± SEM.

**Figure 6 cells-11-02540-f006:**
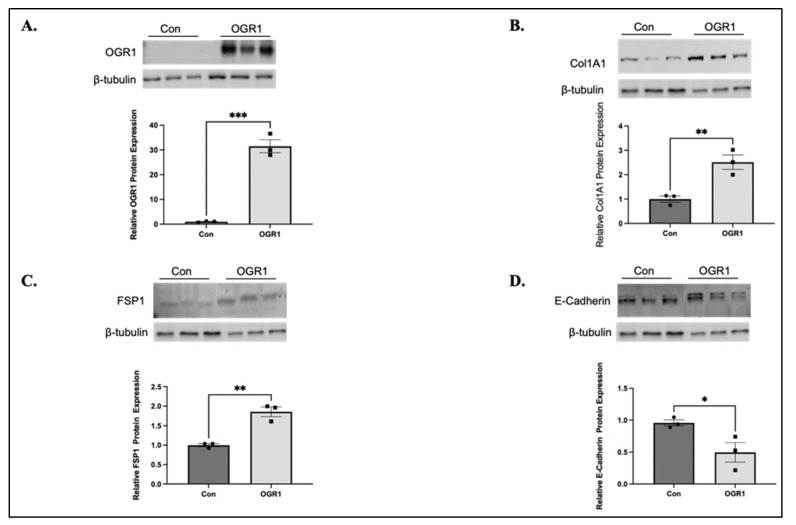
Overexpression of OGR1 in 16-HBE cells causes EMT. We then examined the effects of OGR1 overexpression in 16-HBE cells under basal conditions. OGR1 was successfully transfected into 16-HBE cells (**A**, *** *p* = 0.0003). We then assessed for changes in protein expression that are consistent with EMT. OGR1 overexpression led to significant increases in collagen 1A1 (**B**, ** *p* = 0.0092), fibroblast specific protein 1 (FSP1) (**C**, ** *p* = 0.0028), and decreased E-cadherin expression (**D**, * *p* = 0.0440). Data represent mean expression ± SEM.

**Figure 7 cells-11-02540-f007:**
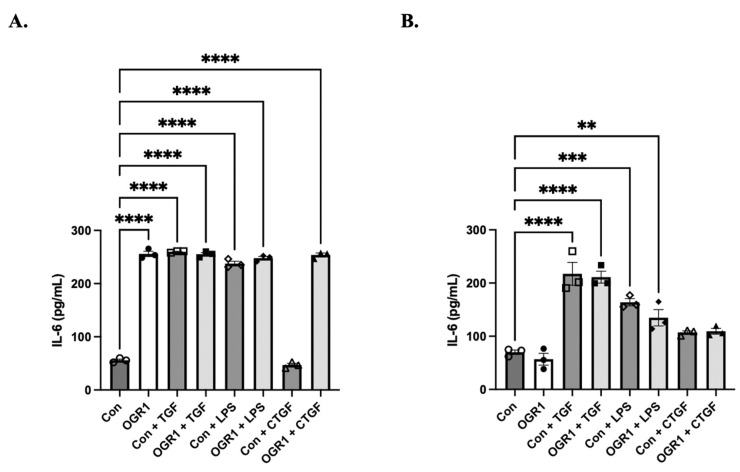
OGR1 is pro-inflammatory in 16-HBE but not in healthy human fibroblasts. Here, 16-HBE cells were treated with either control (GFP) or OGR1 plasmids followed by treatment with TGF-β, LPS, or CTGF, and secreted IL-6 levels were measured via ELISA. In 16-HBE cells, OGR overexpression alone caused significant IL-6 secretion (**A**, **** *p* < 0.0001). As expected, TGF-β, LPS, and CTGF also induced significant IL-6 secretion (**A**, **** *p* < 0.0001 for all significant treatment conditions). However, we did not observe an additive or synergistic effect with OGR1 and other cytokines/growth factors. In contrast, OGR1 overexpression did not induce IL-6 secretion in fibroblasts (**B**). However, TGF-β and LPS did cause significant increases in secreted IL-6 (**B**, **** *p* < 0.0001, *** *p* = 0.0002, ** *p* = 0.0057), whereas CTGF did not. Interestingly, OGR1 overexpression appears to have no effect on inflammatory signaling in fibroblasts. Open circles represent basal control conditions and closed circles represent OGR1 under basal conditions. Open boxes represent control plasmid plus TGF-β, and closed boxes represent OGR1 plasmid plus TGF-β. Open diamonds represent control plasmid plus LPS, and closed diamonds represent OGR1 plasmid plus LPS. Open upward arrows represent control plasmid plus CTGF, and closed upward arrows represent OGR1 plasmid plus CTGF.

**Figure 8 cells-11-02540-f008:**
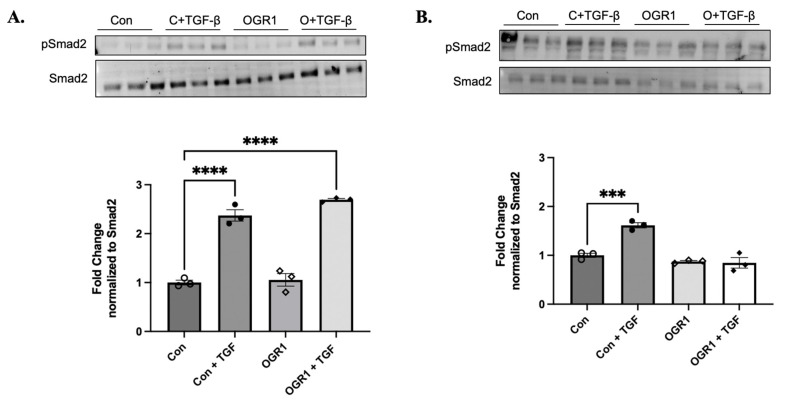
OGR1 does not inhibit Smad2 phosphorylation in 16-HBEs but does so in fibroblasts. Here, 16-HBE cells were transfected with either GFP or OGR1 plasmids and subsequently treated with TGF-β. Canonical TGF-β signaling was then assessed by western blot, examining changes in phosphorylated Smad2 relative to total Smad2. Although TGF-β significantly increased phosphorylated levels of Smad2 (**A**, **** *p* < 0.0001), overexpression of OGR1 was unable to mitigate TGF-β-induced Smad2 phosphorylation (**A**, **** *p* < 0.0001). Fibroblasts underwent identical transfection and treatment, and TGF-β significantly increased Smad2 phosphorylation (**B**, *** *p* = 0.0007). However, in the presence of OGR1 overexpression, TGF-β was unable to phosphorylate Smad2 (**B**, difference between Con and OGR1 + TGF-β, *p* = 0.0031 by Student’s *t*-test). Data represent mean expression ± SEM. Open circles represent control plasmid and closed circles represent control plasmid plus TGF-β. Open diamonds represent OGR1 plasmid and closed diamonds represent OGR1 plasmid plus TGF-β.

**Figure 9 cells-11-02540-f009:**
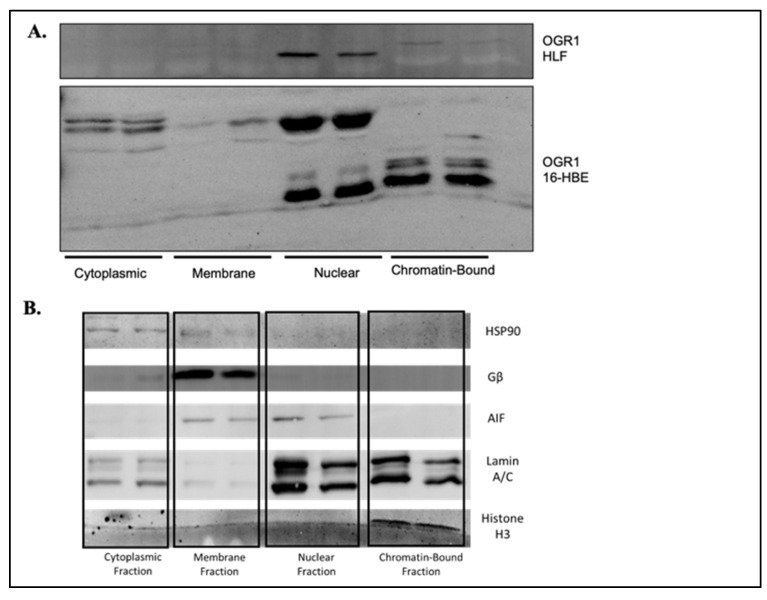
OGR1 has cell-type-specific subcellular localization. Fibroblasts and 16-HBE cells were subjected to subcellular fractionation, and western blot was performed to assess where OGR1 was located. Representative western blots are displayed with duplicate samples per fraction. In fibroblasts, OGR1 was selectively located in the nuclear fraction (**A**, upper panel). However, in 16-HBE cells, OGR1 was in the cytoplasmic, nuclear, and chromatin-bound fractions; to a lesser degree, OGR1 was in the membrane-bound fraction (**A**, lower panel). Treatment with TGF-β did not cause translocation to different subcellular fractions (data not shown). The purity of subcellular fractions was verified by the markers shown (**B**). The unique location of OGR1 in different cell types may explain some of the functional and signaling differences observed between 16-HBE and fibroblasts.

## Data Availability

Not applicable.

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
