# Peer review of "Cell-Type-Specific Effects of the Ovarian Cancer G-Protein Coupled Receptor (OGR1) on Inflammation and Fibrosis; Potential Implications for Idiopathic Pulmonary Fibrosis"

_cells, 2022, doi:10.3390/cells11162540_

Round 1
Reviewer 1 Report
The authors have uncovered a novel role for the proton-sensing GPCR OGR1 (aka GPR68) in the development of idiopathic pulmonary fibrosis (IDF). They show that OGR1 is differentially expressed at the protein level (not mRNA) in lung tissue isolated from IDF patients (decreased) versus normal lung tissue. They show that OGR1 expression is negatively regulated by TGF-β1 (same authors have previously shown that TGF-β1 is activated by low pH and promotes myofibroblast differentiation). Further compelling evidence demonstrate that OGR1 inhibits myofibroblast differentiation and does not contribute to inflammation, OGR1 promotes EMT and inflammation. The authors performed subcellular fractionation for both cell types and show differential subcellular localization dependent on cell type where OGR1 is expressed in the cytoplasm, nuclear fraction and chromatin of epithelial cells, and localized to the nuclear fraction only in fibroblasts – subcellular localization may help explain the differential effects of OGR1 i.e., promoting inflammation in epithelial cells but not in fibroblasts.
This is a clear, well-written manuscript, logical rationale for pursuing proton-sensing GPCRs in IDF and connecting previous findings to current findings to reveal a novel role for OGR1/GPR68. Strengthening the novelty of. thier findings, the authors show that OGR1 exhibits differential subcellular localization of OGR1 that appears to be cell-type specific adding to a growing list of GPCRs localizing to the nucleus.
Critique
Major
1. Given that OGR1 is a GPCR and it is known that low pH results in its constitutive activity promoting Gs and Gq signaling (OGR1 also reported to signal through G11/12) - this begs the question, which OGR1-mediated G protein signaling pathway contributes to the differential effect of OGR1 in each cell type - this could be interrogated using the commercially available reagents such as Gq inhibitor YM254890 or PKA inhibitors to assess the contribution of Gq and Gs signaling respectively to the cellular responses i.e., IL6 secretion, SMAD phosphorylation, myofibroblast differentiation
Minor
2. Line 60 -61. When the other proton-sensing GPCRs are introduced, can the authors, as they did for OGR1 (aka GPR68), please include other names that these receptors are also known as i.e., TDAG aka GPR65; G2A aka GPR132 etc.
3. MW markers or MW sizes are lacking in WB containing figures as well as the original data/images need to be labelled including MW markers.
4. To be consistent all 'OGR' should read 'OGR1' throughout the manuscript, abstract and some figure legends included.
Author Response
Critique
Major
- Given that OGR1 is a GPCR and it is known that low pH results in its constitutive activity promoting Gs and Gq signaling (OGR1 also reported to signal through G11/12) - this begs the question, which OGR1-mediated G protein signaling pathway contributes to the differential effect of OGR1 in each cell type - this could be interrogated using the commercially available reagents such as Gq inhibitor YM254890 or PKA inhibitors to assess the contribution of Gq and Gs signaling respectively to the cellular responses i.e., IL6 secretion, SMAD phosphorylation, myofibroblast differentiation
We appreciate Reviewer 1’s comments. Previous studies in 16-HBE cells and intestinal epithelial cells demonstrate that OGR1 primarily utilizes Gq signaling (references included in the manuscript). We recently demonstrated that dermal, orbital, and pulmonary fibroblasts utilize OGR1 via Gs signaling. This was demonstrated using an OGR1-specific allosteric modulator, Ogerin, and PKA-dependent signaling was enhanced in low pH (6.8 vs 7.4) environments. However, signaling was completely mitigated using H-89 as a pre-treatment (accepted at PLOS One last week, PONE-D-22-03653R1). The discussion section has been updated to include this information and we feel this sufficiently addresses the reviewer’s concerns without the need for additional experimentation.
- Line 60 -61. When the other proton-sensing GPCRs are introduced, can the authors, as they did for OGR1 (aka GPR68), please include other names that these receptors are also known as i.e., TDAG aka GPR65; G2A aka GPR132 etc.
We updated this information.
- MW markers or MW sizes are lacking in WB containing figures as well as the original data/images need to be labelled including MW markers.
These details have been added to the original images, however, we did not do the same for the main manuscript. We feel that this doesn’t add useful information to the reader and clutters the presentation, especially because the images are cropped. We feel this is consistent with many basic science manuscripts published today. If the reviewer prefers, we can add the molecular weight in parentheses after the protein of interest label.
- To be consistent all 'OGR' should read 'OGR1' throughout the manuscript, abstract and some figure legends included.
The draft has been updated to reflect this suggestion.
Reviewer 2 Report
In this manuscript, the authors showed the role of OGR1 receptor in the pathogenesis of IPF. The manuscript is well written and the authors presented their work and findings in an acceptable way. Here are some possible suggestions for the authors that might improve the quality of their manuscript.
1-In the current work was performed only on fibroblasts or epithelial cells, which did not mimic the complex cell-cell interactions occurring in vivo. It would be more reliable if they could also test the effect of OCR1 overexpression/knockdown on the healthy/IPF lung tissue using the ex-vivo method {Precision cut lung slices}.
2-Demonstration of the statistical significance in figures is confusing and needs to be simplified:
-Since the authors compare all the experimental groups to the control, there is no need to draw a line between the target group and the control group. They can just put the statistical significance indicator over the group column.
-No need to add "ns" to indicate there is no significant difference.
-It is the first time to see 4 astersks as an indicator for statistical significance. 3 astersks should be used, even if the p value was less than 0.0001.
Author Response
In this manuscript, the authors showed the role of OGR1 receptor in the pathogenesis of IPF. The manuscript is well written and the authors presented their work and findings in an acceptable way. Here are some possible suggestions for the authors that might improve the quality of their manuscript.
1-In the current work was performed only on fibroblasts or epithelial cells, which did not mimic the complex cell-cell interactions occurring in vivo. It would be more reliable if they could also test the effect of OCR1 overexpression/knockdown on the healthy/IPF lung tissue using the ex-vivo method {Precision cut lung slices}.
We agree with Reviewer #2 that this represents a weakness of the manuscript. While it would be informative to do similar experiments in an ex vivo system, we do not currently have that capability in our laboratory. However, we do have preliminary data (that is not ready to include in this manuscript due to its relative immaturity) that does support OGR1 as being inflammatory in vivo. Briefly, using a small molecule positive allosteric modulator of OGR1 (Ogerin), given to C57Bl6 mice via oropharyngeal aspiration, we found that mice that received both bleomycin and Ogerin died at even more accelerated rates compared to bleomycin alone. BALF analysis demonstrated that OGR1 stimulation induced neutrophilic inflammation, suggesting that OGR1 is pro-inflammatory in vivo.
2-Demonstration of the statistical significance in figures is confusing and needs to be simplified:
-Since the authors compare all the experimental groups to the control, there is no need to draw a line between the target group and the control group. They can just put the statistical significance indicator over the group column.
The statistical analysis is presented according to the default setting of the latest version of GraphPad Prism (9). Unfortunately, when attempting to remove the line indicators, it erases the statistical notation as well. We feel that by removing the “ns”, it presents the data in a clear and concise manner.
-No need to add "ns" to indicate there is no significant difference.
We have removed the “ns” from the graphs
-It is the first time to see 4 astersks as an indicator for statistical significance. 3 astersks should be used, even if the p value was less than 0.0001.
This is the default display for this version of GraphPad Prism (9), we did not alter the way the program automatically displays the information. This program designates p < 0.0001 with 4 asterisks, likely to differentiate from less statistically significant values.
Round 2
Reviewer 2 Report
The authors answered my questions in a good and convincing way
Author Response
Thank you for the opportunity to respond to your comments.
Sincerely,
David